Genome-wide identification of CaARR-Bs transcription factor gene family in pepper and their expression patterns under salinity stress

Alshegaihi Rana M. rmalshegaihi@uj.edu.sa
Alshamrani Salha Mesfer
Department of Biology, College of Science, University of Jeddah , Jeddah , Saudi Arabia
Azeem Farrukh
Electronic publication date: 2023 Nov 1
Publication date: 2023
Volume: 11
Electronic Location ID: e16332
Received 2023 Mar 30; Accepted 2023 Sep 30
Copyright: © 2023 Alshegaihi and Alshamrani
Copyright year: 2023
Copyright holder: Alshegaihi and Alshamrani
License: This is an open access article distributed under the terms of the Creative Commons Attribution License, which permits unrestricted use, distribution, reproduction and adaptation in any medium and for any purpose provided that it is properly attributed. For attribution, the original author(s), title, publication source (PeerJ) and either DOI or URL of the article must be cited.
License URL: https://creativecommons.org/licenses/by/4.0/

Keywords: Capsicum annuum, CaARRs-type B transcription factor, Gene structure, Phylogenetic analysis, Expression analysis, Salinity stress

Funding: This research study received no external funding.

==============================
In plants, ARRs-B transcription factors play a crucial role in regulating cytokinin signal transduction, abiotic stress resistance, and plant development. A number of adverse environmental conditions have caused severe losses for the pepper (Capsicum annuum L.)—a significant and economically important vegetable. Among the transcription factors of the type B-ARRs family, multiple members have different functions. In pepper, only a few members of the ARRs-B family have been reported and characterized. The current study aimed to characterize ARRs-B transcription factors in C. annuum, including phylogenetic relationships, gene structures, protein motif arrangement, and RT-qPCR expression analyses and their role in salinity stress. In total, ten genes encode CaARRs-B transcription factors (CaARR1 to CaARR10) from the largest subfamily of type-B ARRs were identified in C. annum. The genome-wide analyses of the CaARRs-B family in C. annuum were performed based on the reported ARRs-B genes in Arabidopsis. An analysis of homologous alignments of candidate genes, including their phylogenetic relationships, gene structures, conserved domains, and qPCR expression profiles, was conducted. In comparison with other plant ARRs-B proteins, CaARRs-B proteins showed gene conservation and potentially specialized functions. In addition, tissue-specific expression profiles showed that CaARRs-B genes were differentially expressed, suggesting functionally divergent. CaARRs-B proteins had a typical conserved domain, including AAR-like (pfam: PF00072) and Myb DNA binding (pfam: PF00249) domains. Ten of the CaARRs-B genes were asymmetrically mapped on seven chromosomes in Pepper. Additionally, the phylogenetic tree of CaARRs-B genes from C. annuum and other plant species revealed that CaARRs-B genes were classified into four clusters, which may have evolved conservatively. Further, using quantitative real-time qRT-PCR, the study assessed the expression patterns of CaARRs-B genes in Capsicum annuum seedlings subjected to salt stress. The study used quantitative real-time qRT-PCR to examine CaARRs-B gene expression in Capsicum annuum seedlings under salt stress. Roots exhibited elevated expression of CaARR2 and CaARR9, while leaves showed decreased expression for CaARR3, CaARR4, CaARR7, and CaARR8. Notably, no amplification was observed for CaARR10. This research sheds light on the roles of CaARRs-B genes in pepper’s response to salinity stress. These findings enrich our comprehension of the functional implications of CaARRs-B genes in pepper, especially in responding to salinity stress, laying a solid groundwork for subsequent in-depth studies and applications in the growth and development of Capsicum annuum.

Introduction

Peppers (Capsicum annuum L.) are important and widely grown vegetables belonging to the family Solanaceae, which includes eggplants, tomatoes, potatoes, etc. In temperate and subtropical regions worldwide, C. annuum is commonly cultivated as a seasoning vegetable. Peppers distribution, growth, and development are influenced by abiotic stresses such as drought, high salinity, and temperature extremes. Consequently, it is imperative to understand the mechanisms that contribute to pepper’s tolerance of such conditions (Chen et al., 2015; Naaz et al., 2023).

Cytokinins are adenine derivatives, which are important plant hormones that affect nearly every aspect of plant development and growth. Among their functions, they promote chloroplast growth, delay senescence and regulate shoot and root development (Mok & Mokeds, 1994; Haberer & Kieber, 2002; Kakimoto, 2003; Werner et al., 2003). Signals are transduced using two-component phosphorelay systems, and elements of these systems act in response to cytokinin and ethylene hormones, as well as to red light and osmosis (Schaller, 2000; Hutchison & Kieber, 2002; Hwang, Chen & Sheen, 2002; Mason et al., 2005). Initially, the two-component systems were discovered in bacteria, which are controlled by His sensors kinase and response regulators (Mizuno, 1997; Stock, Robinson & Goudreau, 2000). A close correlation has also been found between endogenous cytokinin levels and pleiotropic developmental dysregulation (Ferreira & Kieber, 2005). A two-component signaling pathway is mediated by CKs through histidine kinases (HKs) and response regulators (RRs).

There are three types of Arabidopsis response regulators according to their domain structure and sequence, and referred to as type-A and type-B, (Imamura et al., 1999), and a more recently reported type-C that have the same domain structure as type-A RRs, but they are not induced by cytokinin (Mizuno, 2004). Numerous studies indicate that type-B ARRs contain DNA binding and receiver domains (Sakai, Aoyama & Oka, 2000; Lohrmann et al., 2001; Hosoda et al., 2002). Based on their core receiver domain structures and C-terminal domain sequences, Arabidopsis’ genome contains 23 authentic response regulators (ARRs), divided into four types: A, B, C, and pseudo (Schaller et al., 2007). DNA-binding and receiver domains distinguish Type-B ARRs as transcription factors (Lohrmann et al., 2001; Hosoda et al., 2002). There is compelling evidence that type-A ARR genes are direct targets of phosphorylated ARR-B regulators (Hwang, Chen & Sheen, 2002; Imamura et al., 2003). The ARR-Bs have also been shown to positively regulate cytokinin signaling (Mizuno, 2004). In this study, the type-B authentic response regulator is unified as ARR-Bs. ARRs are phosphorylated by HKs upon perception of a stress signal, leading to changes in their DNA-binding activity and subsequent regulation of target genes. Several studies have demonstrated the involvement of ARRs in plant responses to salt stress. For instance, in Arabidopsis thaliana, ARR2 and ARR12 were found to be upregulated in response to salt stress, and loss-of-function mutants of ARR2 displayed increased sensitivity to salt stress, indicating the importance of ARR2 in salt stress tolerance (Mizuno, 2004). Similarly, in rice, the overexpression of OsRR22, a member of the ARR family, enhanced salt stress tolerance by upregulating the expression of stress-responsive genes (Zhang et al., 2019).

Lately, the pepper genome has been sequenced (Kim et al., 2014; Qin et al., 2014; Magdy et al., 2019; Magdy & Ouyang, 2020). A wide range of RNA molecules derived from several tissues, including root, shoot, leaf, flower, and fruit are also available. Using these data sets, pepper improvement and basic research can identify and functionalize a gene family from a global perspective. The purpose of this study was to identify all potential ARR-B genes encoded in the pepper genome. Routine bioinformatic approaches were conducted to examine the chromosomal distribution, phylogenetic relationships, and gene structure of the identified CaARR-Bs genes. Additionally, functional predictions were made based on gene expression analysis across various developmental stages and in response to salinity stress. These comprehensive results serve as a crucial basis for future investigations focused on gene family exploration and the functional characterization of ARR-Bs in pepper plants.

Materials and Methods

Identification of members of the CaARR-Bs gene family in pepper

Using The Arabidopsis Information Resource (TAIR) database (https://www.arabidopsis.org/), we were able to retrieve Arabidopsis thaliana structural domains for ARR-Bs genes. The obtained sequences were used as probes for homology searches using BLASTp (Wang et al., 2020). BLASTP searches selected the sequences of CaARR-Bs genes to the online pepper genomics database (https://solgenomics.net/organism/Capsicum_annuum/genome), where the genome database of the C. annuum cultivar Znula was selected (Qin et al., 2014). The search yielded pepper ARR-Bs candidate genes. The physicochemical properties of the ARR-Bs family genes were estimated using Expasy (http://web.expasy.org/) (Gasteiger et al., 2005). Finally, the subcellular localization analysis of the pepper CaARR-Bs gene family was performed using Cell-Ploc 2.0 (http://www.csbio.sjtu.edu.cn/bioinf/Cell-PLoc-2/). The CaARR-Bs gene family protein sequences were aligned with A. thaliana ARR-B genes by MAFFT aligner (Katoh & Standley, 2013) using the embedded algorithms in Geneious Prime. Subsequently, the rooted phylogenetic trees were generated using the maximum likelihood methods (ML). The ML tree was constructed using FastTree V2 (Kumar et al., 2018), embedded in Geneious Prime.

Chromosomal location, gene structure, domain annotation, and secondary structure analysis

The chromosomal location of CaARR-Bs genes throughout the pepper genome was investigated using the MG2C V2 online tool (Chao et al., 2021). GSDS was used to identify CaARR-Bs gene exon-intron structures by comparing the genomic sequences vs cDNA sequences (http://gsds.cbi.pku.edu.cn/). The CaARR-Bs and the Myb DNA binding domains (a domain that distinguishes the ARR-A from ARR-B genes) were detected using CDD (Marchler-Bauer et al., 2015), SMART, pfam and InterProScan databases (Wang et al., 2022). SPOMA was used to predict CaARR-Bs secondary structures (Sapay, Guermeur & Deléage, 2006).

Potential cis-element analysis in promoter regions

A BLASTn search was conducted on the pepper genome using the cultivar Znula as the query sequence to identify CaARR-Bs genes. For each gene, a 1,500 bp upstream of the initiation codon was retrieved and analyzed for cis-elements using PlantCARE database (http://bioinformatics.psb.ugent.be/webtools/plantcare/html/).

Plant materials and salinity stress treatment

Seeds of the pepper cultivar Gedeon F1 (Capsicum annuum L.; https://www.syngenta.com/en) were sterilized with 1% sodium hypochlorite for 30 min, washed with sterile water, and then sown in perlite beds at 28 °C (Qin et al., 2014). Seedlings were grown under 16 h/light at 25 °C and 8 h/night at 18 °C with a relative humidity of 60% until they had six leaves. Plants were irrigated with Hoagland solution at half-strength pH 5.6. Leaves and roots were harvested from the seedlings with three biological replicates as control. For the salt stress experiment, five-week-old plants were irrigated with 200 mM NaCl (Wang et al., 2022). After 12 and 24 h of treatment, each sample (3–4 leaves) was collected in three biological replicates. Liquid nitrogen was used to rapidly freeze samples, followed by −80 °C storage until RNA extraction.

RNA extraction and qPCR analysis

Total RNA was extracted from pepper tissues (leaves and roots) subjected to different salt stress treatments using the EasyPure® Plant RNA Kit (TransGen Biotech, Beijing, China) following the instructions provided by the kit manufacturer. The quality and quantity of the extracted RNA were evaluated using electrophoresis on 2% agarose gels and a Quantus™ Fluorometer (Promega, Madison, WI, USA). The cDNA synthesis was performed using SuperScript III reverse transcriptase (Invitrogen, Carlsbad, CA, USA) and adjusted to a concentration of 100 ng µL−1. During the PCR amplification, a range of 108 to 147 base pairs was targeted while avoiding the conserved region (Table 1).

Table 1 New primer list designed to quantify CaARRs-B genes in pepper.

The list includes details on the position start (min) and end (max) of each primer, along with the sequence and expected size in base pair (bp).

GeneID	Direction	Min	Max	5′-Seq-3′	Exp. size (bp)	
CaARR1	F	1,282	1,301	TCTAATCATGTCGCCCCAGC	133	
R	1,395	1,414	GCTTCCAGTGCCAAGTCTCT	
CaARR2	F	738	757	ACTGATGAACGTTCCCGGAC	147	
R	865	884	CAAAAGGTTGTGTCTGGGCG	
CaARR3	F	1,307	1,326	TTTCTCAGCCTCCGTTGTCC	128	
R	1,415	1,434	AGATGCTGGGGAGACTGGAT	
CaARR4	F	813	832	AGTTCAACAACAGGGTGGCA	148	
R	941	960	CTCGGCATGAAGAGCTGTCA	
CaARR5	F	1,212	1,231	CATGGCCTTCCCGACCTATC	143	
R	1,335	1,354	AATTCTCGGGTGGTTGCCAT	
CaARR6	F	980	999	TTCGCAACCTGACAGCATCT	147	
R	1,107	1,126	ATGCGACGTGGACAATGACT	
CaARR7	F	774	793	AGTCGCAAGCCATCTTCAGA	110	
R	864	883	TGACATAGTGCCCTGGAGGA	
CaARR8	F	832	851	GCTGCTGCATTAGGGGGTAA	108	
R	920	939	CTGACCCTGACCGAACCTTC	
CaARR9	F	333	352	AAGCAGGGTGATGAAGGGTG	133	
R	446	465	ATTTCCAACGTCCCTTGCCT	
CaARR10	F	571	590	CGTGTACTTTGGTCACCGGA	145	
R	696	715	TCTGAAGGTGGCTAGCAACG	

Quantitative real-time RT-PCR (qRT-PCR) was performed using TransStart® Green qPCR SuperMix (TransGen Biotech, Beijing, China). The amplification reactions were performed following: 95 °C for 5 min, followed by 40 cycles of 95 °C for 15 s, 55 °C for 20 s, and 72 °C for 30 s. Melting curve analysis, performed by increasing the temperature from 55 to 95 °C (0.5 °C per 10 s), and a gel electrophoresis of the amplified fragments confirmed that the product contained single amplicons. In each experiment, the relative fold differences were calculated using the ΔΔ Ct. Normalization was conducted using GAPDH Ct values, amplified using 5′-ATGATGATGTGAAAGCAGCG-3′ and 5′-TTTCAACTGGTGGCTGCTAC-3′ as a reference gene (Arce-Rodríguez & Ochoa-Alejo, 2015). In this experiment, three biological replicates per sample were used. A statistical student’s t-test was used to compare gene expression levels.

Results

Identification of the CaARR-Bs family genes members in pepper

The identification of the CaARR-Bs family genes in the C. annuum genome (cultivar Gedeon F1) based on the pepper genome sequences database was performed using BLASTp database search to query Arabidopsis thaliana ARR-Bs genes. In total, ten sequences were putative as pepper ARR-Bs genes (CaARRs). The chromosome location, exon number, and genomic and physiochemical characteristics of each gene are in Table 2. In the current study, the shortest putative open reading frame (ORF) was 1,374 pb, while the longest was 2,025 pb, with the amino acid length of CaARRs proteins ranging from 389 to 684 pb, a molecular weight (Mw) of 43.09 to 75.16 kDa, and a theoretical isoelectric point (PI) varying from 5.39 to 9.47. In addition, the presence of the ARR-like domain was verified using the Pfam database, and its position was unfixed among all copies, while all copies were localized at the subcellular level in the nucleus.

Table 2 List of CaARRs-B family genes identified in pepper includes genomic and physiochemical characteristics of each gene.

Name	Gene ID	Genomic location	ORFa	AAb	pId	Mwe	ARR-Bc	Localization predicted	
CaARR1	Capana01g000809	Chr01:16748515:16754075: +	1737	579	5.70	64.34	20–126	Nucleus	
CaARR2	Capana01g002340	Chr01:148777192:148781202: −	2013	671	5.72	74.35	34–142	Nucleus	
CaARR3	Capana05g000373	Chr05:8121795:8127559: +	1992	664	5.39	73.47	32–140	Nucleus	
CaARR4	Capana05g000907	Chr05:35121771:35129633: −	1908	636	6.17	69.38	28–136	Nucleus	
CaARR5	Capana06g001571	Chr06:38165007:38169434: −	1674	558	6.19	61.80	20–127	Nucleus	
CaARR6	Capana07g000239	Chr07:10638104:10640334: −	1167	389	9.48	43.09	22–131	Nucleus	
CaARR7	Capana07g000240	Chr07:18343788:18352137: +	2010	670	6.76	73.59	22–136	Nucleus	
CaARR8	Capana09g000064	Chr09:1430274:1434744: +	2052	684	5.84	75.16	26–134	Nucleus	
CaARR9	Capana11g001030	Chr11:98813212:98816708: −	1347	449	7.42	50.05	27–136	Nucleus	
CaARR10	Capana12g000157	Chr12:2698224:2702038: +	1695	565	5.61	63.33	23–130	Nucleus	
Notes:

a Open reading frame (bp).

b Amino acid.

c ARR-B domain.

d Theoretical isoelectric point.

e Molecular weight (kDa).

Alignment and phylogenetic analysis of CaARR-Bs

Phylogenetic analysis was conducted to confirm the identified CaARRs copies and determine the evolutionary affinity of the CaARR-Bs genes and ARR-Bs genes of Arabidopsis. With full-length amino acid sequences, the alignment of the ARR-BS domain was conducted and used to perform an unrooted phylogenetic tree with a bootstrap test. The analyzed ARR-Bs proteins were grouped into five distinct clusters. The CaARR-Bs proteins in pepper were clustered into four of the five subclusters with strong bootstrap support (Fig. 1). Functional divergence could have resulted from the presence or absence of species-specific CaARR-Bs.

Figure 1 (A) Alignment of the ARR domain sequences from 10 putative ARRs-B genes in pepper and 21 ARRs-B genes from Arabidopsis thaliana. ARRs-B and Myb DNA binding motifs on amino acid sites are marked at the top, and sequence identities are shown below. (B) An unrooted phylogenetic tree displays CaARRs-B genes’ relationships in C. annuum and A. thaliana. Different colors indicate the five different groups.

Numbers at nodes represent bootstrap values based on 1,000 replicates.

Chromosomal location and duplication event of CaARR-Bs genes

As indicated by the starting and ending positions of CaARR-Bs genes on the chromosomes in Table 1, the genomic DNA sequences of each CaARR-Bs gene were mapped to the chromosomal location (Fig. 2). With few exceptions, CaARR-Bs genes were mainly found at the extremities of their respective chromosomes. The CaARR-Bs genes were asymmetrically distributed on chromosomes 1, 5, 6, 7, 9, 11, and 12, and none were found on chromosomes 2, 3, 4, 8, and 10. The CaARR1 and 2 genes were located on chromosome 1, CaARR3 and 4 were located on chromosome 5 with a separation of 10 Mb, and CaARR5 was on chromosome 6. In addition, CaARR6 and 7 located on chromosome 7 with a separation of <1 Mb, while chromosome 9 had CaARR8, chromosome 11 had CaARR9, and CaARR10 was in chromosome 12 (Fig. 2).

Figure 2 Chromosomal distribution of CaARRs-B genegenes in pepper.

Chromosomal distribution of CaARRs-B genegenes in pepper. The scale is in million bases (Mb). Chromosomes without CaARRs genes are not shown.

Predicted secondary structures of CaARR-Bs proteins

The prediction of the secondary structure of CaARR-Bs proteins was conducted. The primary constituent forms of the secondary structure of the pepper ARR-Bs protein, with α-helices, β-turns, and extended chains, were investigated and measured. According to structural predictions, pepper ARR-Bs gene family members contain α-helices, irregularly coiled, extended chains, and β-turns. The most average proportion for alpha-helices was around 34.1 ± 8%; the highest percentage was from CaARR6 (84.7%) and the lowest was from CaARR2 (26.3%). Most genes have averaged around 21.6 ± 4% β-turns; the highest percentage was from CaARR3 (26.3%) and the lowest was from CaARR5 (13.8%). The results show that the irregular coiling average record was 23.3 ± 6%, as the highest proportion was for CaARR7 (30.3%), while the lowest was for CaARR6 (13.7%; Table 3).

Table 3 Predicted secondary structure of the CaARRs-B family proteins.

GeneID	Alpha helix (%)	Beta strand (%)	Coil (%)	Turn (%)	
CaARR1	28.8	19.1	24.0	28.1	
CaARR2	26.3	24.0	27.3	22.4	
CaARR3	32.9	26.2	18.7	22.1	
CaARR4	34.2	21.4	27.9	16.4	
CaARR5	43.3	13.8	20.8	22.1	
CaARR6	48.7	18.5	13.7	19.2	
CaARR7	26.6	22.7	30.3	20.3	
CaARR8	27.5	20.6	29.6	22.4	
CaARR9	40.3	24.3	17.8	17.6	
CaARR10	31.9	25.0	23.2	20.0	
Average ± SD	34.1 ± 8	21.6 ± 4	23.3 ± 6	21.1 ± 3	

Gene structure of CaARR-Bs proteins

For each gene in the CaARR-Bs gene family, exon-intron distribution and conserved motifs were analyzed. For each gene, several isoforms were detected with the exception for CaARR2 and CaARRs 5–10 (unique isoform). The CaARR1 recorded two isoforms where one was complete and a shorter isoform (x2: 1,431 bp and 477 aa). The CaARR3 recorded three isoforms: one complete and two shorter isoforms (x2: 1,992 bp and 664 aa; x3: 1,956 bp and 652 aa). Finally, the CaARR4 recorded five isoforms where one was complete, and five shorter isoforms (x2: 1,785 bp and 595 aa; x3: 1,896 bp and 632 aa; x4: 1,899 bp and 633 aa; x5: 1,908 bp and 636 aa).

There were differences in exonic and intronic regions between the 10 CaARR-Bs. Exons ranged from 5 to 11 while introns ranged from 4 to 10. The CaARR-Bs genes were clustered according to their sequence homology into three groups, showing similar gene structures within each group. Regardless of their chromosomal location, one cluster included CaARR6 and CaARR8, another included CaARR7, CaARR1, and CaARR5, and the other included CaARR2, CaARR4, and CaARR9 (Fig. 3).

Figure 3 A schematic diagram of the 10 CaARRs-B gene structures showing exons and introns structures.

Potential cis-element analysis in promoter regions of CaARR-Bs genes

To further characterize the potential regulatory mechanisms of CaARR-Bs, 1,500 bp upstream sequences from the translation start sites were analyzed to ensure the inclusion of potential enhancers (they can be located up to 1 Mbp away from the gene, but typically, they are found within a shorter distance, around 50–1,500 bp, upstream or downstream from the start site; Pennacchio et al., 2013). Based on their biological significance, cis-elements in CaARR-Bs were identified and categorized. The promoter sequences of 10 CaARR-Bs genes contained 146 possible cis-elements. Three main groups of cis-acting elements were identified: hormones, environmental stress, and photoresponses. The CaARR-Bs promoters included transcription factors binding sites related to abscisic acid responsiveness, auxin responsiveness, defense and stress responsiveness, endosperm expression, enhancer-like involved in anoxic specific inducibility, light responsiveness, low temperature responsiveness, meristem expression, zein metabolism regulation, MeJA responsiveness, Gibberellin responsiveness, ATBP 1 transcription factor, MYB binding site involved in drought inducibility, MYB binding site involved in flavonoid biosynthetic genes regulation, MYB binding site involved in light responsiveness, MYBHv1 binding site, and Salicylic acid responsiveness (Fig. 4). According to the cis-component, CaARR-Bs genes can respond to different abiotic stresses.

Figure 4 The number of various cis-elements on the promoters of each CaARRs-B gene.

Promoter sequences (−1,500 bp) of 10 CaARRs-B genes were analyzed.

CaARR-Bs gene expression profiles in response to salinity stress

The expression level of the CaARR-Bs was measured with qRT-PCR for seedlings exposed to salt stress. The expression patterns of CaARR-Bs genes varied among stress levels and showed considerable variation in expression patterns of the CaARR-Bs genes over time in the roots of plants more than in leaves. A heat map was generated to show the expression profiles while the expression differences were color-indicated (Fig. 5). In response to salinity stress, the expression of patterns of CaARR2 and CaARR9 in the roots were up-regulated after 12 and 24 h; additionally, CaARR5 and CaARR6 were up-regulated in roots after 24 and 12 h, respectively. Meanwhile, the CaARR3, CaARR4, CaARR7, and CaARR8 were highly expressed in control samples and were down-regulated after 12 and 24 h of treatments. In the leaves, the expression level of CaARR1 was the only expressed CaARR gene, and only in control samples, followed by CaARR5, which showed slight expression after 24 h of salinity treatment. The expression of CaARR2, CaARR6, and CaARR9 increased slightly after 24 h of salinity treatment. The expression of CaARR3, CaARR4, CaARR7, and CaARR8 was down-regulated in 24 h. The expression level on the leaves of CaARR6, CaARR5, and CaARR67 was decreased, while CaARR3 was induced after 12 and 24 h of stress. The CaARR10 was the only copy that showed no amplification curves during the qPCR.

Figure 5 Expression profiles of nine pepper ARRs-B genes in different tissues.

Expression profiles of 9 pepper ARRs-B genes in different tissues, a case-oriented PCA based on the complete qPCR profile (A), and a gene-oriented heatmap (B) generated using the heat mapper tool were shown. Blue, white, and red colors correspond to low, moderate, and high fold change levels. Z-score is the normalization for the heatmap values.

Discussion

The His-Asp phosphorelay signaling pathway in prokaryotic cells is controlled by ARRs (Suzuki et al., 1998). On the basis of their phylogeny, the ARR-B gene family evolved from nonvascular plants (bryophytes) such as phylostratum, while several orthologous genes exist in the plants (Cuming et al., 2007). The ARR-B genes have exhibited evolutionary conservation across all the selected species. This conservation is evident in their expansion within higher plants. ARRs have been classified as types A and B based on their conserved domains (Imamura et al., 1999). An N-terminal DNA-binding and receiver domains indicate that ARR-B functions as a transcription factor (Ishida et al., 2008). A key component of cytokinin signaling pathways, type B-ARRs genes have been linked to plant response to various environmental stresses; in response to different stress conditions, superoxide anion and hydrogen peroxide contents were measured as antioxidant enzyme activity (Nakamichi et al., 2009). There are ARR-Bs transcription factors identified in Arabidopsis genomes (Mason et al., 2004; Ramírez-Carvajal, Morse & Davis, 2008), rice (Schaller et al., 2007), peach (Zeng et al., 2017), pear (Ni et al., 2017), soybean (He et al., 2022), and fragrant rice (Rehman et al., 2022). The number of ARR-Bs genes varied among plant species, ranging from a single copy (green algae) to 63 copies (Populus trichocarpa) with an average of 14.3 ± 11.2 copies per plant species (PlantTFDB V4; Jin et al., 2017). Such variation in copy numbers can be attributed to factors such as gene duplication events, evolutionary divergence, functional diversification, and selective pressures (Choi et al., 2014). Despite this, little is known about ARR-Bs in pepper. Studies have shown that gene organization plays an important role in how multiple gene families evolve (Xu et al., 2012; Ullah et al., 2019; Wang et al., 2020; Arce-Rodríguez, Martínez & Ochoa-Alejo, 2021; Ahiakpa et al., 2022). In the current study, ten CaARR-Bs genes in pepper have been identified and characterized by their structure, cis-elements in the promoter regions, chromosomal location, gene duplication, and phylogeny. In addition, salinity stress affects the expression profile of different tissues. Thus, in this study, the CaARR-Bs genes were comprehensively analyzed to determine their biological function.

From the public genomic data, we derived the genomic sequences, protein sequences, and chromosomal locations of the identified CaARR-Bs genes (Chao et al., 2021). Following the standard protocol, the secondary structures of CaARR-Bs proteins were putatively predicted (Sapay, Guermeur & Deléage, 2006) and functionally characterized with proper tools (Wang et al., 2022). The 10 predicted CaARR-Bs genes have been identified on seven chromosomes, indicating a segmental repetition of the gene family. During plant genome evolution, duplication or large-scale segmental duplication is thought to produce gene families (Cannon et al., 2004). Many transcription factor families have been reported in gene duplication events, including HD-ZIP, C2H2-ZF, SlARR-B, MYB, and NAC (Liu et al., 2015; Chen et al., 2015; Arce-Rodríguez, Martínez & Ochoa-Alejo, 2021).

All the CaARR-Bs homologous gene pairs identified using the pepper genome vs the Arabidopsis genome showed tight phylogenetic clustering; their topologies are more closely related, suggesting they are more closely related. In addition, intron numbers were related to CaARR-Bs gene classifications. The duplication of genes is crucial to genomic expansion and realignment (Kumar, Tyagi & Sharma, 2011). Based on the phylogenetic tree and synteny analysis, the results were consistent. Genetic evolution has been attributed to gene or genome duplication events as the primary source of variation in the CaARR-Bs family gene. The result showed that segmental duplication events promote the evolution of CaARR-Bs genes (Yang, Tuskan & Cheng, 2006; Yang et al., 2008; Gillis et al., 2009).

Further clarifying the roles of CaARR-Bs promoter regions in response to abiotic stresses, we also identified several conserved cis-regulatory elements. An analysis of cis-acting elements in pepper type-B ARRs genes showed a close relationship between these genes and growth, hormonal signal transduction, and abiotic stress resistance. Several cis-elements involved in drought resistance are found in the promoter region of ARR-B, and a triple mutant lacking all three of these genes was reported in Arabidopsis (Nguyen et al., 2016). Further, ARR-B belongs to the helix-loop-helix family and are nuclear-localized transcription factors, as evidenced by their helix-loop-helix structure in the CaARR-Bs domain (Hosoda et al., 2002). The ARR-Bs regulators target type-A ARRs genes directly, which are activated by phosphorylated ARR-Bs (Hwang & Sheen, 2001; Imamura et al., 2003).

In Capsicum, the involvement of two-component response regulators (ARRs) has been identified as key components in these signaling pathways. Salt-responsive genes are regulated by ARRs, which mediate the plant’s adaptive response. The involvement of ARRs in salt stress signaling pathways can be attributed to their ability to regulate downstream stress-responsive genes. ARRs function by interacting with other transcription factors and cis-acting elements in the promoter regions of target genes, thereby modulating their expression. This regulatory mechanism enables ARRs to orchestrate the activation of stress-responsive genes involved in ion homeostasis, osmotic regulation, antioxidative defense, and other adaptive processes that contribute to salt stress tolerance in Capsicum (Urao, Yamaguchi-Shinozaki & Shinozaki, 2000).

There is a close relationship between the expression profiles of genes and their biological functions. Despite this, ARR-B expression patterns in different tissues have rarely been studied in Pepper. Studies have shown that ARR-B transcription factors are involved in cytokinin signal transduction; in addition, they may play a role in root development and drought- and salinity-tolerance (Garay-Arroyo et al., 2012; Kiryushkin et al., 2019; Seo et al., 2020; Alwutayd et al., 2023; Abd El-Moneim et al., 2015, 2022). Based on the qPCR analysis, we found that these genes had broad expression profiles in Pepper, CaARR2, CaARR5, CaARR6, and CaARR9 were highly expressed in the roots. while CaARR2, CaARR4, CaARR6, and CaARR9 were down-regulated in the shoots. Interestingly, the Arabidopsis ortholog of SlARR-B1 regulates sodium accumulation in tomato shoots (Mason et al., 2010). Several gene families, including ARR-B, have been identified and their expression profiles have been characterized according to the stress responses and phytohormone responses of different plant species (Ye et al., 2009; Zhu et al., 2013; Zhao et al., 2016; Xia et al., 2017; Wang et al., 2017; Liu et al., 2020; Zhang et al., 2020; He et al., 2020). These studies provide a broader context for our findings on ARR-B expression in Pepper, highlighting the complex interplay of gene expression, stress responses, and phytohormone signaling in plant species.

Conclusion

To conduct functional studies, it is essential first to characterize and classify gene families. During the present study, 10 CaARR-Bs genes were identified and classified. Seven of the twelve chromosomes of C. annuum contained uneven distributions of genes. According to the phylogenetic analysis, most CaARR-Bs presented possible orthologs in Arabidopsis, indicating a common evolutionary origin. Moreover, different salinity stress levels induced different expression levels of CaARR-Bs genes in leaves and roots, supporting the theory that CaARR-Bs have functionally divergent functions. By integrating our results, we identified CaARR-Bs candidates that might contribute to regulating salt stress resistance and shed new light on CaARR-Bs transcription factors’ role in secondary metabolism. To better understand how CaARR-Bs function and how they are regulated in Capsicum spp., more interspecific functional characterization of CaARR-Bs genes is required.

Supplemental Information

Supplemental Information 1 CaARR-B qPCR Calculations.

Click here for additional data file.

Additional Information and Declarations

Competing Interests

Author Contributions

Data Availability

The authors declare that they have no competing interests.

Rana M. Alshegaihi conceived and designed the experiments, performed the experiments, analyzed the data, prepared figures and/or tables, authored or reviewed drafts of the article, and approved the final draft.

Salha Mesfer Alshamrani conceived and designed the experiments, performed the experiments, analyzed the data, prepared figures and/or tables, authored or reviewed drafts of the article, and approved the final draft.

The following information was supplied regarding data availability:

The raw data is available in the Supplemental File.

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
