# Peer review of "Genome-wide identification of CaARR-Bs transcription factor gene family in pepper and their expression patterns under salinity stress"

_PeerJ, doi:10.7717/peerj.16332_

## Round 0.1 · original submission · Major Revisions

The reviewers have suggested major revisions for your manuscript. You are requested to address all the comments.

Reviewer 1 ·

Basic reporting

The manuscript entitled with “Genome-wide identiûcation and expression analysis of CaARRs-B transcription factor gene family in pepper (Capsicum annuum L.) under salinity stress” by Alshegaihi et al provide some knowledge of pepper ARR-B family. The gene number, distribution, structure, and protein properties of CaARR genes were analyzed, and their expression patterns in response to salt stress were analyzed. This manuscript contributes to the knowledge of ARR-B gene family in pepper. However, it lacks novelty, and some further experiments are required. The major comments are listed as follows.
1. The language is not good, and there are many grammar mistakes. So, an extensive language processing is required.
2. The title is not accurate. “Genome-wide identification of CaARRs-B transcription factor gene family in pepper and their expression patterns under salinity stress” might be better.
3. The abstract, the language is not concise. For example, in lines 7-8, the sentence “phylogenetic relationships, gene structures, protein motif arrangement, and RT-qPCR expression” is repeat with the sentence in lines 13-14.
4. In the introduction, some simple introductions of CKs and ARR-B were included. the roles of CKs and ARRs in the stress responses, especially in salt stress responses are lacked. And also, are there some reports about the functions of pepper ARR-Bs? Because the authors said that “In pepper, only a few members of the ARRs-B family have been reported and characterized. in the abstract section.

Experimental design

5. Materials and Methods: The ARR-Bs in tomato have been identified (Wang J, Xia J, Song Q, Liao X, Gao Y, Zheng F, Yang C. Genome-wide identification, genomic organization and expression profiles of SlARR-B gene family in tomato. J Appl Genet. 2020, 61(3):391-404. doi: 10.1007/s13353-020-00565-5.). The authors should use tomato ARR-Bs as queries in blast and other bioinformatic analysis. Page 8 line 97, what is “GSDS”?

Validity of the findings

6. Results section:
(1) The subtitles should be corrected.
(2) In 3.4 section, there are many transcript isoforms for each CaARR, however, whether these transcripts really exist? Some RT-PCR experiments should be performed to confirm these transcripts.
(3) The tissue expression patterns of CaARRs should be performed.
(4) Why 12h and 24h of salt treatment were choosed?
7. Discussion section:
The authors used two paragraphs to discuss the roles of segmental duplication events in the CaARR gene duplication however, in the results section, no corresponding analysis (such as KaKs calculation) was performed.
(2) Why and how CaARRs are involved in salt stress repsonses, it is not discussed.
8. The conclusion section:
In Line 299, the authors said that “shed new light on CaARRs-B transcription factors’ role in secondary metabolism”. However, no result was found about the secondary metabolism.

Reviewer 2 ·

Basic reporting

The authors performed a genome-wide analysis of the ARR-B transcription factor family in Capsicum annuum and investigated the expression profiles of these genes under salinity stress. This research is a good foundation that can be used for further research involving functional validation studies. Therefore, I recommend that the manuscript be accepted for publication after completion of major revisions.
The following changes can be made to improve the quality of the grammar used throughout the manuscript:
• Lines 15-17: needs restructuring
• Line 23: replace with “functional divergence”
• Lines 28-30: needs restructuring
• Lines 32-34: could be worded better
• Lines 74-76: needs restructuring
• Line 80: should it be “The Arabidopsis thaliana sequences…”
• Line 99: The citation of Marchler-Bauer et al 2015 is for the NCBI CDD, not the GSDS database that was discussed in that line.
• All the instances of “cis” in ”cis-element” needs to be checked for italicization
• Line 104: sentence needs rewording
• Line 111: “28º C following” needs checking
• Lines 123-125: recheck
• Lines 143-144: The units for some of the quantities (e.g. MW) are missing.
• Lines 151-153: This line looks like it should be put in the methods section, especially since there was no mention of these programs in the methods section.
• Line 165: replace the phrase “take place”
• Line 173: “… and the lowest…”
• Line 212: I don’t see any highlighting of significant differences in Figure 5
• Lines 232-235: needs restructuring
• Line 249: The citation of Marchler-Bauer et al 2015 is for the NCBI CDD, not SMART
• Lines 247-251: These belong in the methods section and not discussion
• Line 264: “family genes”
• Line 278: “Despite this, ARR-B expression patterns in different tissues have rarely been detected in Pepper.” Didn’t you detect expression of these genes? This sentence needs clarifying.
• Line 284: “downregulated on in”
• All instances of the words downregulated and upregulated should be hyphenated (down-regulated, up-regulated)
• Line 297: “… have functionally…”
• Throughout the manuscript, the gene family is referred to as the ARRs-B family, while in the discussion, it is referred to as the ARR-Bs. The latter seems more grammatically correct.

Experimental design

The methods look experimentally sound and all underlying datasets have been provided. The following points can be used to improve the manuscript.
• In the manuscript, GAPDH is listed as the housekeeping gene. Actin is listed as the housekeeping gene in the Supplementary file. Please explain this discrepancy.
• The primers for the housekeeping gene are missing from Table 1.
• Was there z-score normalization of the heatmap values? Is so, this can be included in the figure legend.

Validity of the findings

• The authors referred to the expression values as fold changes, although the data calculations in Supplementary File 1 only show calculations for relative expression values. The findings could be made more robust by calculating the fold change of the 12H sample expression relative to that of the control (and similarly, for the 24H samples). An appropriate statistical test could be used to determine whether there are statistically significant differences in gene expression. This would allow the author’s claims of significant expression changes (Line 212) to be more scientifically sound.
• Lines 253-254: “… suggesting the family consists of segmental repetitions”. How does the location of the genes on 7 chromosomes suggest this? If this information was presented in previous literature, then it needs to be cited. Otherwise, I find it hard to accept this statement.
• Lines 30-32: “In particular, the expression of CaARR3, CaARR5, CaARR6, CaARR7, and CaARR8 showed significant expression levels in roots.” The title of this manuscript deals specifically with the roles of ARR-B genes during salinity stress. If a researcher wanted to conduct further research on this topic, they would want to see in the abstract at a glance which of these genes were the most likely to be involved in the pepper salinity stress response, yet no mention was made of such genes in the abstract or conclusion. If the suggested fold change calculations (see point 1 of this section) were performed, these results could be presented as part of the abstract.

Additional comments

no comment

---

## Round 0.2 · Minor Revisions

The manuscript is significantly improved. I have included a few minor suggestions for your consideration (file attached). You are requested to address the reviewer's comments and my suggestions to revise the current version of the manuscript.

Reviewer 2 ·

Basic reporting

The requested changes were made. A few more points to note:
• Lines 70 and 88: italicize Arabidopsis thaliana
• Line 73: Italicize gene name
• Line 107: distinguish should be plural
• Line 112: a 1500 bp sequence upstream…
• Line 118: the degrees Celsius symbol (in 28º C) isn’t written like the other times you mentioned it correctly (e.g. 25 °C)
• Line 131: the -1 in µL-1 should be written in superscript
• Lines 151-152: units are missing
• Line 185: Gene structure “of"? Also, it’s gene structure you’re looking at, so it can’t be of the ARR-B proteins like you mentioned, but of the genes
• Line 255: sentence needs rewording

Experimental design

The requested changes were made.

Validity of the findings

• The required changes were made. If student’s t-test was used to look for significant differences in gene expression, this could be mentioned in the Methods section.
• In the abstract, you should talk about the genes that were significantly up-regulated in roots and/or leaves during salinity stress, rather than whether they were highly expressed in roots (line 30). If I were studying pepper plants, I would want to know at a glance (by reading your abstract), which would be the best genes for future functional validation studies on their roles in salinity stress. You mention CaARR-B3, -7, and -8 in the abstract, even though its expression is down-regulated under salinity stress in the root.

Additional comments

no comment

---

## Author Rebuttal · Round 0.2

**Dear Editor / Reviewer**

**We would like to thank the Editor and the reviewers for the valuable comments and constructive suggestions to our manuscript, which greatly helped to enhance the quality of our manuscript.**

## Reviewer 1

### Basic reporting

The manuscript entitled with "Genome-wide identiûcation and expression analysis of CaARRs-B transcription factor gene family in pepper (*Capsicum annuum* L.) under salinity stress" by Alshegaihi et al provide some knowledge of pepper ARR-B family. The gene number, distribution, structure, and protein properties of CaARR genes were analyzed, and their expression patterns in response to salt stress were analyzed. This manuscript contributes to the knowledge of ARR-B gene family in pepper. However, it lacks novelty, and some further experiments are required. The major comments are listed as follows:

1. The language is not good, and there are many grammar mistakes. So, extensive language processing is required.

**Response:** Thank you for your comment, a language service will be consulted to revise the final version of the manuscript.

2. The title is not accurate. "Genome-wide identification of CaARRs-B transcription factor gene family in pepper and their expression patterns under salinity stress" might be better.

**Response:** We have no problem with changing the title to the suggested one, thank you.

3. The abstract, the language is not concise. For example, in lines 7-8, the sentence "phylogenetic relationships, gene structures, protein motif arrangement, and RT-qPCR expression" is repeat with the sentence in lines 13-14.

**Response:** We are grateful for the observation; the abstract was rephrased considering such notes.

4. In the introduction, some simple introductions of CKs and ARR-B were included. the roles of CKs and ARRs in the stress responses, especially in salt stress responses are

lacked. And also, are there some reports about the functions of pepper ARR-Bs? Because the authors said that "In pepper, only a few members of the ARRs-B family have been reported and characterized. in the abstract section.

**Response:** Thank you very much, the introduction was rewritten to cover the raised concerns.

**Experimental design**

5. Materials and Methods: The ARR-Bs in tomato have been identified (Wang J, Xia J, Song Q, Liao X, Gao Y, Zheng F, Yang C. Genome-wide identification, genomic organization and expression profiles of SlARR-B gene family in tomato. J Appl Genet. 2020, 61(3):391-404. doi: 10.1007/s13353-020-00565-5.). The authors should use tomato ARR-Bs as queries in blast and other bioinformatic analysis. Page 8 line 97, what is "GSDS"?

**Response:** We appreciate your comment and suggestion. Effectively we have used the tomato first and yielded discrepancies in terms of names and similarity. Thus, we followed and presented the same approach followed by Wang et al. (2020), and other similar studies to ensure the clearness of the outputs.
The "GSDS" is the name of the server used to visualize the genes features (Introns/exons) of the studies genes. It stands for Gene Structure Display Server.

**Validity of the findings**

6. Results section:

(1) The subtitles should be corrected.

**Response:** Revised and corrected.

(2) In 3.4 section, there are many transcript isoforms for each CaARR, however, whether these transcripts really exist? Some RT-PCR experiments should be performed to confirm these transcripts.

**Response:** All the detected copies were quantified with qPCR under control and salinity conditions, while only the CaARR10 was not active at neither condition, as shown in point 3.7.

(3) The tissue expression patterns of CaARRs should be performed.

**Response:** We have performed the qPCR of the CaARRs in two different tissues, the roots vs the leaves, as shown in point 3.7.

(4) Why 12h and 24h of salt treatment were chosen?

**Response:** We have performed the salinity treatment experiment, and took zero, 6h, 12h, 24h, 36h and 48h. On one hand, the plant status was severed at 36h and 48h, reflected in a very degraded RNA which would influence the qPCR integrity. On the other hand, the differences between 6h and 12h were null, both showed the same exact effects. Thus, we finally decided to include the control versus the 12h and 24h.

7. Discussion section:

The authors used two paragraphs to discuss the roles of segmental duplication events in the CaARR gene duplication however, in the results section, no corresponding analysis (such as KaKs calculation) was performed.

**Response:** Thank you for your comment, the duplication was suggested as an explanation for having two copies located very close to each other's, in addition to the high similarity in their sequence and expression pattern, which suggest the possible duplication event that may create a gene copy that close, and proximately similar.

(2) Why and how CaARRs are involved in salt stress repsonses, it is not discussed.

**Response:** A paragraph in the discussion was added as suggested.

8. The conclusion section:

In Line 299, the authors said that "shed new light on CaARRs-B transcription factors' role in secondary metabolism". However, no result was found about the secondary metabolism.

**Response:** We consider the role of the CaARRs-B TF in regulating some other genes in response to salinity stress as a significant output of this study, and shed light on their role to regulate substances manufactured by plants that make them competitive in their own environment (Ref: https://www.ncbi.nlm.nih.gov/pmc/articles/PMC7123774/)

# Reviewer 2

## Basic reporting

The authors performed a genome-wide analysis of the ARR-B transcription factor family in Capsicum annuum and investigated the expression profiles of these genes under salinity stress. This research is a good foundation that can be used for further research involving functional validation studies. Therefore, I recommend that the manuscript be accepted for publication after completion of major revisions. The following changes can be made to improve the quality of the grammar used throughout the manuscript:

• Lines 15-17: needs restructuring

**Response:** Restructured and rephrased as suggested.

• Line 23: replace with "functional divergence"

**Response:** Changed as suggested.

• Lines 28-30: needs restructuring

**Response:** Restructured and rephrased as suggested.

• Lines 32-34: could be worded better

**Response:** Rephrased as suggested.

• Lines 74-76: needs restructuring

**Response:** Restructured and rephrased as suggested.

• Line 80: should it be "The Arabidopsis thaliana sequences…"

**Response:** "sequences" was added to the sentence.

• Line 99: The citation of Marchler-Bauer et al 2015 is for the NCBI CDD, not the GSDS database that was discussed in that line.

**Response:** Thank you for the observation, we have added the missing information related to the citation. Which might be deleted by accident during the revision of the manuscript before submission.

• All the instances of "cis" in "cis-element" needs to be checked for italicization

**Response:** We rechecked all the instances and all were italicized if wasn't.

• Line 104: sentence needs rewording

**Response:** Rephrased as suggested.

• Line 111: "28º C following" needs checking

**Response:** Rechecked and corrected as suggested.

• Lines 123-125: recheck

**Response:** Rechecked and corrected as suggested.

• Lines 143-144: The units for some of the quantities (e.g., MW) are missing.

**Response:** kDa unit was added to Mw numbers, we checked the other quantities for the same comment.

• Lines 151-153: This line looks like it should be put in the methods section, especially since there was no mention of these programs in the methods section.

**Response:** We have replaced the given information in methods section and rephrased it, as suggested.

• Line 165: replace the phrase "take place"

**Response:** Replaced with located.

• Line 173: "… and the lowest…"

**Response:** Corrected.

• Line 212: I don't see any highlighting of significant differences in Figure 5

**Response:** The sentence was rephrased in accordance with the figure.

• Lines 232-235: needs restructuring

**Response:** Rechecked and corrected as suggested.

• Line 249: The citation of Marchler-Bauer et al 2015 is for the NCBI CDD, not SMART

**Response:** Noted and a missing sentence was added.

• Lines 247-251: These belong in the methods section and not discussion

**Response:** Rechecked and rephrased to be part of the discussion.

• Line 264: "family genes"

**Response:** Corrected to "gene family".

• Line 278: "Despite this, ARR-B expression patterns in different tissues have rarely been detected in Pepper." Didn't you detect expression of these genes? This sentence needs clarifying.

**Response:** We meant "rarely studied". The sentence was corrected in this sense.

• Line 284: "downregulated on in"

**Response:** Corrected from "on" to "in".

• All instances of the words downregulated and upregulated should be hyphenated (down-regulated, up-regulated)

**Response:** Corrected as suggested.

• Line 297: "… have functionally…"

**Response:** Corrected as indicated.

• Throughout the manuscript, the gene family is referred to as the ARRs-B family, while in the discussion, it is referred to as the ARR-Bs. The latter seems more grammatically correct.

**Response:** We changed all to the *ARR-Bs.*

**Experimental design**

The methods look experimentally sound and all underlying datasets have been provided. The following points can be used to improve the manuscript:

• In the manuscript, GAPDH is listed as the housekeeping gene. Actin is listed as the housekeeping gene in the Supplementary file. Please explain this discrepancy.

**Response:** GAPDH was the HKG used for this study, we regret the mistake in the Supp file, mostly due to using older template to calculate the fold change values.

• The primers for the housekeeping gene are missing from Table 1.

**Response:** Primer sequences for the GAPDH were added to the text.

• Was there z-score normalization of the heatmap values? Is so, this can be included in the figure legend.

**Response:** Yes, it was standardized using z-score. The information was added to the figure legend as suggested.

**Validity of the findings**

• The authors referred to the expression values as fold changes, although the data calculations in Supplementary File 1 only show calculations for relative expression values. The findings could be made more robust by calculating the fold change of the 12H sample expression relative to that of the control (and similarly, for the 24H samples). An appropriate statistical test could be used to determine whether there are statistically significant differences in gene expression. This would allow the author's claims of significant expression changes (Line 212) to be more scientifically sound.

**Response:** We appreciate the suggestion. We compared the value using student t-test between both tissues; it seems 02, 03, 04, 06, 07, 08 and 09 showed significant difference of expression in one tissue versus the other. We are not sure if including the cases per tissue in one test is correct or you may suggest a different model. We would not add to the text without your approval.

| | CaARR 01 | CaARR 02 | CaARR 03 | CaARR 04 | CaARR 05 | CaARR 06 | CaARR 07 | CaARR 08 | CaARR 09 | CaARR 10 |
|---|---|---|---|---|---|---|---|---|---|---|
| Tissue | 0.237875 | **0.000878** | **0.049697** | **0.045947** | 0.228873 | **0.021109** | **0.026414** | **0.026414** | **0.012113** | NA |

• Lines 253-254: "… suggesting the family consists of segmental repetitions". How does the location of the genes on 7 chromosomes suggest this? If this information was presented in previous literature, then it needs to be cited. Otherwise, I find it hard to accept this statement.

**Response:** We are referring to the cytological explanation for the observed gene copy repetition at a proximate distance <1 kbp. We don't hold for as an evident fact, rather an explanation for the observed adjacent locations of two copies of the same gene family.

• Lines 30-32: "In particular, the expression of CaARR3, CaARR5, CaARR6, CaARR7, and CaARR8 showed significant expression levels in roots." The title of this manuscript deals specifically with the roles of ARR-B genes during salinity stress. If a researcher wanted to conduct further research on this topic, they would want to see in the abstract at a glance which of these genes were the most likely to be involved in the pepper salinity stress response, yet no mention was made of such genes in the abstract or conclusion. If the suggested fold change calculations (see point 1 of this section) were performed, these results could be presented as part of the abstract.

**Response:** We totally agree, the abstract will be refined accordingly in the next round after agreeing on the statistical design and comparison scheme you may see fit. Thank you very much for your time and effort.

---

## Round 0.3 · accepted · Accept

The authors have significantly improved the manuscript. It can now be accepted for publication.